# High-Risk Node-Positive Hormone Receptor-Positive/HER2-Low Breast Cancer Relapse on Adjuvant Abemaciclib Treatment with ER Loss at Metastatic Recurrence: A Case Report and Literature Review

**DOI:** 10.3390/diagnostics15233042

**Published:** 2025-11-28

**Authors:** Maria Vita Sanò, Lorenza Marino, Maria Puleo, Sarah Pafumi, Stefano Marletta, Giuseppina Rosaria Rita Ricciardi, Carlo Carnaghi

**Affiliations:** 1Department of Medical Oncology, Humanitas Istituto Clinico Catanese, 95045 Catania, Italy; lorenza.marino@humanitascatania.it (L.M.); maria.puleo95@gmail.com (M.P.); sarah.pafumi@humanitascania.it (S.P.); stefano.marletta@humanitascatania.it (S.M.); carlo.carnaghi@huamanitascatania.it (C.C.); 2Department of Molecular Medicine, Sapienza University of Rome, 00185 Rome, Italy; 3Department of Onco-Haematology, Papardo Hospital, 98100 Messina, Italy; giusyricciardi81@hotmail.it

**Keywords:** metastatic breast cancer, adjuvant CDK4/6 inhibitors, estrogenic receptor loss, PI3KCA mutation, breast cancer relapse, triple-negative conversion, trastuzumab-deruxtecan, Her2low, ADC, case report

## Abstract

**Background and Clinical Significance**: Up to 30% of patients with HR+/HER2-early breast cancer (eBC) may experience distant recurrence, and patients with high-risk clinical features have a higher likelihood of recurrence. For these patients, the addition of a CDK4/6 inhibitor to standard adjuvant endocrine therapy (ET) has demonstrated a significant reduction in the risk of invasive and distant recurrence. Despite that, a small subset of patients still experience distant relapse. To date, the characteristics of patients who relapse on adjuvant CDK 4/6i are not well defined. **Case Presentation**: Here we report the case of a patient with early-stage HR+/HER2− breast cancer at high risk of recurrence, who experienced distant metastatic relapse during adjuvant therapy with abemaciclib plus ET, with a shift in the tumor subtype to triple-negative. Genomic alterations associated with ET and cyclin-dependent kinase 4 and 6 inhibitor resistance were analyzed with next-generation sequencing (NGS) carried out at recurrence. Results showed P53 and PI3KCA pathway alterations, but no ESR1 mutation or RB1 mutations. The duration was 12 months for adjuvant abemaciclib, and the first-line metastatic treatment lasted less than 3 months. Conversely, T-DXd administered due to the presence of HER2-low showed excellent effectiveness. **Conclusions**: The management of breast cancer relapse occurring during adjuvant therapy with CDK4/6 inhibitors is challenging. Performing a re-biopsy is crucial due to the possible loss of estrogen receptors, which would require a change in therapeutic strategy no longer based on endocrine therapy. In cases that remain luminal, knowledge of the mutational profile may help to offer patients novel targeted treatments.

## 1. Introduction

Breast cancer is the most common cancer and the second leading cause of cancer-related mortality among women worldwide. More than 90% of patients with breast cancer are diagnosed at an early stage (eBC), of whom approximately 70% have hormone receptor-positive (HR+) and human epidermal growth factor receptor 2-negative (HER2−) tumors [1]. Adjuvant endocrine therapy (ET) improves outcomes for patients with HR+/HER2− eBC and is the standard of care (SOC) in this setting [2]. However, many patients experience disease recurrence [3]. Approximately 50% of recurrences occur within 5 years of diagnosis and continue beyond 20 years even with SOC adjuvant ET [4,5,6,7]. Up to 30% of patients with HR+/HER2− eBC may experience distant recurrence, which is incurable, and patients with high-risk clinical and/or pathological features have a higher likelihood of recurrence, particularly during the initial years of adjuvant endocrine therapy (ET) [8,9,10,11,12]. For these patients, the addition of a CDK4/6 inhibitor (abemaciclib or ribociclib) to standard adjuvant ET has demonstrated a significant reduction in the risk of invasive and distant recurrence. The monarchE trial assessed 2 years of adjuvant abemaciclib in combination with ET vs. ET alone in patients with high-risk node-positive HR+/HER2− eBC [13]. The NATALEE trial evaluated the addition of 3 years of ribociclib to SOC adjuvant nonsteroidal aromatase inhibitor (NSAI) vs. NSAI alone in a broad population of patients with stage II/III HR+/HER2− eBC with high-risk node-negative (N0) or node-positive disease [10]. In the adjuvant setting, both CDK4/6i (ribociclib and abemaciclib) have shown statistically significant improvements in invasive disease-free survival (iDFS) in patients with HR+/HER2− eBC. Furthermore, abemaciclib has shown an overall survival (OS) benefit, whereas OS data for ribociclib remain immature. Despite that, a small subset of patients still experience distant relapse, either during or shortly after completing the adjuvant CDK4/6i regimen [13,14]. At present, identifying the optimal therapeutic strategy for patients who experience relapse after or during adjuvant CDK4/6i plus ET remains a challenge, as this population is underrepresented in clinical trials. For example, only approximately 2% of such patients were included in the recent postMONARCH and INAVO120 and 1% in CAPITELLO 291 trials [15,16,17]. Moreover, there is a lack of data regarding the efficacy of ADCs or oral SERDs in this clinical setting. In addition, to date, the characteristics of patients who relapse on adjuvant CDK4/6i are not well defined. Herein, we report a case of a high-risk eBC HR+/HER2− relapsed on adjuvant abemaciclib treatment with conversion to triple-negative mBC. This case confirms the importance of performing a biopsy at disease recurrence during adjuvant ET and abemaciclib, as conversion to triple-negative radically changes the pharmacological treatment. Additionally, it shows an excellent response to the ADC trastuzumab deruxtecan in a case of conversion from luminal breast cancer to triple-negative during adjuvant abemaciclib and ET, a scenario for which we currently have no available data to guide therapeutic algorithms.

## 2. Case Report

A 48-year-old premenopausal woman, with no relevant comorbidities and a family history of breast cancer (paternal aunt), presented in August 2022 with a self-detected palpable mass in the left breast. Mammography and ultrasound revealed an irregular mass (~4.5 cm) with nipple retraction and lymphadenopathy axillary lymph node metastases. Core needle biopsy confirmed invasive carcinoma of no special type (NST), grade 2, ER 90%, PgR 90%, Ki-67 40%, HER2 2+ (FISH negative), both on the primary tumor and the lymph node metastasis. Magnetic Resonance Imaging revealed multicentric disease; staging CT (computer tomography) and bone scan showed no distant metastases (cT2 cN2 M0, stage IIA AJCC Version 8). Following multidisciplinary assessment, the patient received neoadjuvant chemotherapy with four cycles of doxorubicin/cyclophosphamide followed by 12 weekly doses of paclitaxel. Breast imaging post-chemotherapy showed a partial radiological response. In May 2023, she underwent skin-sparing left mastectomy with immediate reconstruction and axillary dissection. Histopathology revealed invasive NST, grade 2, ypT2 ypN3a (10/27 nodes), ER 95%, PgR 0%, Ki-67 3%, HER2 2+ (FISH negative), with perineural invasion and no lymphovascular invasion. Her MD Anderson response score was 4.173 (class II). She underwent BRCA testing, which was found to be wild-type. In July 2023, she started adjuvant therapy with triptorelin, letrozole, and abemaciclib. Moreover, in June 2023, post-mastectomy radiation therapy (PMRT) was delivered to the reconstructed breast and regional lymphatics. In August 2024, after 12 months of adjuvant treatment with ET and abemaciclib, the patient reported concern about the appearance of erythematous skin with nodular lesions on the reconstructed breast. Additionally, she had developed back pain for about a month, which prevented her from carrying out all daily activities. Suspecting a possible disease recurrence, the patient was offered a skin biopsy and a systemic imaging reassessment.

Punch biopsy of the skin lesions confirmed cutaneous metastases, ER- and PgR-negative, Ki-67 40%, HER2 2+ (FISH negative), consistent with hormone receptor loss. PD-L1 testing (both 22C3 and SP142 tests) was negative. Staging with CT and bone scan revealed lytic lesions in the right sixth rib and lumbar spine (L1–L2). She received palliative radiotherapy on L1–L2 with improvement in pain symptoms, followed by first-line systemic chemotherapy with carboplatin plus gemcitabine. Monthly subcutaneous administration of denosumab was started in September 2024. Assessment of response to chemotherapy performed after 10 weeks showed a progression of cutaneous metastases. Bone disease assessed by TC was stable. In consideration of the presence of HER2-low, the patient was considered for second-line metastatic therapy with T-DXd. The therapeutic choice was made by a multidisciplinary team and discussed with the patient, highlighting the expected benefits and potential toxicities.

Following the start of second-line treatment with trastuzumab deruxtecan in December 2024, a complete remission of the cutaneous metastases was observed, with stable bone disease after 12 weeks. The response was confirmed at the last follow-up in September 2025. The treatment was overall well tolerated, except for moderate asthenia. No alopecia occurred. Grade 1 nausea and vomiting were experienced during the first few days after treatment. The patient, therefore, continues treatment with T-DXd at full dose with preservation of quality of life.

### Circulating Tumor DNA Profiling

At relapse in September 2024, next-generation sequencing (NGS) using the FoundationOne^®^ Liquid CDx assay identified oncogenic mutations in TP53 and PIK3CA. No RB1 or ESR1 alterations were detected, which are typically associated with reduced efficacy of CDK4/6 inhibitors in the metastatic setting or with acquired resistance to endocrine therapy in HR+/HER2-BC (Table 1).

## 3. Discussion

Metastatic recurrence during adjuvant treatment with CDK4/6 inhibitors is not common. Specifically, in the “monarchE” trial, 9.1% of patients experienced early distant recurrence: 6% within 2 years from treatment initiation (planned treatment period) and an additional 3.1% in the third year. In the “NATALEE” trial, 7.1% of patients experienced distant spreading within the first 3 years of ribociclib treatment, while the exact 4-year relapse rates were not reported in the latest trial update [18]. To date, the characteristics of patients who relapse on adjuvant CDK 4/6i are not well defined. Only one small retrospective study on a real-world population evaluated clinicopathological and treatment-related features of patients relapsing during or after adjuvant abemaciclib and ET. Among 163 patients who received adjuvant abemaciclib (from 2018 to 2024), 15 (9.2%) experienced recurrence. The median age at diagnosis was 48.0 years; two patients had germline (g) BRCA2 mutations, and one patient gCHEK2 mutation. Most patients presented with stage II (33.3%) or stage III (53.3%) disease, and the majority had ductal histology (60.0%) and poorly differentiated tumors (46.7%). Pathological T and N categories were predominantly T2 (53.3%) and N2 (46.7%), respectively. Median duration of treatment was 8.0 months for abemaciclib. The median disease-free interval was 19 months, with all recurrences being distant. The liver (40.0%), bone (26.7%), and lung (13.3%) were the most common recurrence sites. Among the 10 patients with available NGS data from recurrence tissue, nearly all exhibited genomic alterations in the P53 pathway. In particular, in this dataset, Corti et al. detected five TP53 oncogenic mutations, one TP53 homozygous deletion, and three MDM2 high amplifications (copy number > 20). One ESR1 oncogenic mutation and no RB1 oncogenic mutations were identified. Additional alterations included those in the phosphatidylinositol-3 kinase pathway (PI3KCA). It is very interesting to note that 50% of these patients showed estrogen receptor loss (ER loss) at recurrence, and the median first-line metastatic treatment lasted 3 months [19].

Prior to the implementation of CDK4/6 inhibitors in the adjuvant setting, ER loss at metastatic relapse was reported in up to 20–30% of luminal-like primary breast tumors [20]. The biopsy of the metastatic site at recurrence, if feasible, is crucial since loss of ER at relapse is not uncommon. The current guidelines recommend offering to perform a biopsy of a metastatic lesion to evaluate receptor status. The choice of systemic therapy in metastatic disease is often based on the receptor status of the primary lesion. As therapeutic decision making is guided by subtype, biopsy of the metastatic lesion to determine receptor status may alter treatment. In the future, maybe the use of 16a-[18F]-fluoro-17b-estradiol (FES) PET/CT could likely be helpful in cases of recurrence in poorly accessible sites. 16α-[18F]-fluoro-17β-estradiol (FES) PET/CT can accurately identify tumor lesions expressing ER and can also predict endocrine therapy (ET) responsiveness in patients with advanced HR+ breast cancer. Since the loss of ER and/or PgR expression in recurrent HR+ disease may arise from clonal selection of biologically heterogeneous primary or metastatic tumor cell populations under the selective pressure of reduced extracellular estrogen levels or pharmacologic inhibition of the ER pathway, repeating FES PET/CT scans during the course of ET may help guide therapeutic decision making [21,22,23,24]. Among the mechanisms responsible for primary or acquired resistance to endocrine therapies is the loss of estrogen and/or progesterone receptor expression, which may render breast cancer cells independent of estrogen stimulation and, consequently, resistant to estrogen deprivation or pharmacological inhibition of estrogen receptors. However, the underlying molecular mechanisms remain unclear. In fact, while several studies have characterized the genomic mechanisms of HR+ BC cell resistance to ETs and their clinical consequences, the molecular mechanisms underlying HR conversion from HR+ to HR-negative (HR) status, as well as their impact on patient prognosis and on the choice of therapeutic strategies, remain much less explored. One major limitation of the molecular studies assessing HR status conversion is the fact that this phenomenon can occur in different phases of BC progression and, as a consequence, it could be driven by different mechanisms.

Specifically, it remains uncertain whether HR status conversion arises spontaneously, results from the selective pressure imposed by estrogen deprivation within the bloodstream or tumor microenvironment, or represents an adaptive mechanism triggered by specific pharmacological agents [25]. Dieci et al. were the first to demonstrate that phenotypic discordance between primary breast cancer and recurrence has a measurable impact on overall survival; among discordant cases, loss of receptor expression was the main factor associated with poorer outcomes [26]. Subsequently, a retrospective analysis of 459 patients with paired primary and recurrent tumor samples revealed that individuals whose tumors converted from HR+ to HR− exhibited a 48% higher risk of death compared with patients whose tumors maintained a stable HR+ status [27]. Although not rare, the clinical presentation, biological behavior, and prognostic implications of these tumors remain poorly characterized. This gap in knowledge is clinically significant, as ER loss is common, yet the most appropriate therapeutic approach for such cases has not been established. Beyond discordance in ER, PR, and HER2 expression by immunohistochemistry, multiple next-generation sequencing (NGS) studies have also documented frequent shifts in intrinsic molecular subtypes in breast cancer [28,29,30,31,32]. Intrinsic molecular subtyping is typically assigned using the PAM50 gene expression-based classifier on RNA sequencing data from breast tumors [33,34]. The intrinsic molecular subtypes are Luminal A, Luminal B, HER2-enriched (HER2-E), basal-like, and normal-like. Some studies have reported that the clinical subtype by IHC does not completely overlap with the intrinsic molecular subtype by NGS, indicating that subtype switching may be more frequent than if only reporting on receptor discordance by IHC [35]. Zhan et al. recently characterized clinical and pathologic features of a subset of ER+/HER2− breast cancer patients who converted to a triple-negative phenotype upon recurrence, and investigated the molecular alterations associated with HR loss during BC progression. Patients were categorized into two groups based on receptor status at local or distant recurrence: concordant ER+/HER2− tumors (*n* = 92) and discordant triple-negative breast cancer (TNBC, *n* = 20). A comparison of the clinicopathologic features of primary tumors revealed similarities in histologic type and grade between the two cohorts. No significant difference in TIL levels was observed between the two groups, suggesting that the immune microenvironment remains relatively stable despite biomarker switching to TNBC at recurrence. All tumor specimens underwent targeted next-generation sequencing (NGS). Discordant TNBC tumors were characterized by a higher prevalence of PTEN mutations (30% vs. 5.5%, *p* = 0.007) compared to concordant ER+/HER2− tumors. These mutations were primarily deleterious, resulting in loss of tumor-suppressor function. In this study authors showed that PTEN inactivating mutations were relatively frequently associated with HR loss in the HR+/HER2− breast cancer recurrence setting, and HR loss demonstrated a stepwise pattern.

Increased signaling via the PI3K/AKT/PTEN pathway may be a mechanism for the transition to hormone independence in recurrent disease [36]. Morganti et al. compared clinicopathological characteristics and clinical outcomes between a cohort of 51 patients with primary ER+/HER2− and paired triple-negative metastasis and two control cohorts of paired early-metastatic ER+/HER2− and triple-negative breast cancers and observed intermediate clinicopathological features and outcomes compared with tumors without receptor conversion at metastatic relapse [37]. Genomic studies showed that the mutational profile of luminal-like and TNBC metastases is largely distinct. Interestingly, PI3KCA alterations, which are frequent among ER-positive tumors, are also reported in ~9% of TNBC. Since up to 40% of androgen receptor (AR)-positive TNBC display activating alterations of the PI3K/AKT pathway [37,38,39], these findings may suggest that at least a subset of secondary TNBC are luminal and that ER-loss does not necessarily correspond to a subtype switch [36,40]. The PI3K/AKT/PTEN pathway is among the most altered signaling pathways in diverse cancer types. Common alterations include activating mutations and/or amplification of PIK3CA, loss of PTEN, and mutation and/or amplification of AKT. Somatic PTEN mutations are identified in 5–10% of breast cancers, and the majority of PTEN gene mutations occur in advanced and metastatic breast cancer [39,41,42]. Loss or reduced PTEN expression is observed in 30–40% of breast cancers and is associated with poor prognosis in breast cancer, particularly in the ER+/HER2− subtype [43,44]. Conversely, high expression of PTEN has been associated with improved response to tamoxifen treatment [45]. PTEN alteration has also been implicated in resistance to endocrine therapy and CDK4/6 and PI3Kα inhibitors [46,47]. Understanding the role of PTEN and its alterations in breast cancer is crucial for developing effective treatment strategies and improving patient outcomes. In our clinical case, the presence of a PIK3CA mutation represents, on one hand, a negative prognostic factor as it is associated with endocrine resistance, and on the other hand, a therapeutic target for PI3K pathway inhibitors in case the patient had not lost hormone receptors at disease recurrence. After HR loss, tumors are treated as ab initio triple-negative breast cancers; however, few studies have described the clinicopathologic features and outcomes of these tumors. Validation of findings on larger cohorts is warranted, and correlative analyses addressing the biology underpinning HR loss are of utmost importance as they could allow for fine-tuning treatments in the early setting, thus avoiding some relapses. Fortunately, not only has the landscape of adjuvant treatment been enriched with new drugs, but so has the metastatic setting, particularly with the introduction of antibody–drug conjugates (ADCs). Among ADCs, trastuzumab-deruxtecan, in the Destiny B-04 trial, significantly improved overall survival (OS) and progression-free survival compared with treatment of the physician’s choice of chemotherapy (TPC) for patients with human epidermal growth factor receptor 2-low (HER2-low) (immunohistochemistry (IHC) 1+ or IHC 2+/in situ hybridization-negative) metastatic breast cancer who had received at least one line of chemotherapy for advanced disease. Median OS also favored T-DXd in exploratory analyses of hormone receptor-negative [48]. Another ADC, Sacituzumab govitecan (SG), a first-in-class anti-trophoblast cell surface antigen 2 (Trop-2) antibody–drug conjugate, demonstrated superior efficacy over single-agent chemotherapy in patients with metastatic triple-negative breast cancer (mTNBC) who have received two or more prior systemic therapies, including at least one of them for advanced disease, in the phase III ASCENT study. Notably, about 30% of the patients were not triple-negative at initial diagnosis, and this discordance between primary tumor staining and metastatic recurrence underscores the importance of obtaining biopsy samples at the time of recurrence, as in our patient. Chemotherapy remains the mainstay of treatment in first-line mTNBC patients who are not candidates for PD-1/PD-L1 inhibitors. However, chemotherapy is associated with low response rates and short progression-free survival, with an overall ~50% of patients do not receive treatment beyond the first-line setting, demonstrating a need for additional effective earlier-line treatment options. In ASCENT-03, SG demonstrated statistically significant and clinically meaningful improvement in PFS compared with treatment of the physician’s choice in patients with first-line metastatic triple-negative breast cancer who are not candidates for checkpoint inhibitors [49]. Finally, Datopotamab-deruxtecan, a TROP2-directed ADC, showed a statistically significant and clinically meaningful improvement in OS compared to chemotherapy as first-line treatment for patients with locally recurrent inoperable or metastatic TNBC who are not eligible for immunotherapy, in the phase III TROPION-Breast02 trial. For the first time, a therapy has demonstrated a significant overall survival benefit over standard chemotherapy in patients who are not candidates for immunotherapy [50]. The results also demonstrated a substantial benefit in progression-free survival (PFS), marking a major advance for this aggressive subtype of breast cancer, where therapeutic options remain limited, and outcomes are poor. Despite recent treatment advances, estrogen receptor-positive (ER+) and HER2-negative (HER2−) advanced breast cancer (aBC) remains a heterogeneous disease with significant unmet need, particularly in patients relapsing early on treatment or shortly after completing adjuvant ET + CDK4/6i. For patients who remain luminal, although clear evidence is lacking, patients relapsing on or after adjuvant CDK4/6i may retain some degree of endocrine sensitivity. Therefore, continuing CDK4/6 inhibition beyond recurrence should not be ruled out, while CT and ADCs could be reserved for patients experiencing early relapse on the adjuvant combination. It is reasonable to assume that treatment decisions may be guided by multiple factors, including disease burden at recurrence, the duration of prior adjuvant CDK4/6i plus ET, and/or the disease-free interval [51]. However, advanced breast cancer remains almost incurable despite significant progress in understanding the disease and increasing availability of different systemic treatment options; eventually, most patients with metastatic breast cancer develop endocrine resistance and progressive disease. Overcoming resistance to endocrine therapy in breast cancer is a major challenge, and there is still an unmet need for safe and efficacious treatment options. Several mechanisms may be responsible for developing endocrine resistance; both PI3K/AKT/mTOR and CDK4/6 are downstream checkpoints of multiple signaling pathways regulating cell growth and survival, and have been demonstrated as relevant targets for new treatment modalities [52,53]. A combination of estrogen receptor (ER) signaling pathway blockades with either PI3K/AKT/mTOR or CDK4/6 inhibition (cell cycle arrest induction) demonstrated significant clinical benefit, as measured by progression-free survival (PFS) in patients with HR+/HER2− advanced and/or metastatic breast cancer [54,55,56]. New potential therapeutic strategies may emerge from ongoing randomized trials, such as PIONERA or CAPItello-292, which are enrolling patients who have relapsed during adjuvant endocrine therapy or within 12 months of its completion, including patients who previously received adjuvant CDK4/6 inhibitors and relapsed at least 12 months after completing treatment. PIONERA will evaluate the efficacy and safety of giredestrant (oral SERD) compared with fulvestrant, both combined with the investigator’s choice of CDK4/6i in participants with ER+ and HER2− aBC who have developed resistance to adjuvant ET. Meanwhile, CAPItello-292 aims to evaluate the efficacy and safety of the added benefit of capivasertib (AKT inhibitor) combined with CDK4/6i and fulvestrant in participants with locally advanced (inoperable) or metastatic HR+/HER2− breast cancer patients who have relapsed during adjuvant endocrine therapy.

## 4. Conclusions

The management of mBC relapse during adjuvant CDK4/6 inhibitors is a challenge, because no data from randomized clinical trials (RCTs) are available. Patients with metastatic triple-negative breast cancer have poor survival outcomes. Although immunotherapy has shown promising first-line clinical activity, for about 70% of patients whose tumors do not express PD-L1—or who cannot receive immunotherapy because of prior exposure, comorbidities, or accessibility issues—chemotherapy remains the first-line standard of care [57]. However, chemotherapy is associated with low response rates and short progression-free survival. Therefore, this group has long faced an urgent need for new, effective options. Given the limited benefit achieved with first-line chemotherapy, ADCs likely represent the best first-line therapy, especially in light of the recent ADC trial efficacy results. Finally, the genomic landscape of relapsed BC following or during adjuvant therapy with CDK4/6i may differ significantly, potentially influencing both treatment choices and sequencing. Patients who remain luminal could allow the use of endocrine-based therapy, while CT and ADCs could be reserved for patients experiencing early relapse on the adjuvant combination. Naturally, our case has limitations; it is a single case with a short follow-up.

However, it emerges that biopsy at recurrence is crucial, as conversion to triple-negative radically changes the therapeutic algorithm; NGS at recurrence can also be important for identifying potential targets for novel targeted therapies. Finally, our case highlights the excellent efficacy of T-DXd in a poor-prognosis setting, where the response to first-line metastatic chemotherapy had been very limited. Its effectiveness, together with the maintenance of quality of life, was essential for implementing the treatment.

## Figures and Tables

**Table 1 diagnostics-15-03042-t001:** FoundationOne^®^ Liquid CDx.

Laboratory Findings: Genomic and Biomarker	
Finding	VAF
PIK3CA E542K	0.64%
TP53 C141Y	0.55%

## Data Availability

The original contributions presented in this study are included in the article. Further inquiries can be directed to the corresponding author.

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
