# Peer review of "High-Risk Node-Positive Hormone Receptor-Positive/HER2-Low Breast Cancer Relapse on Adjuvant Abemaciclib Treatment with ER Loss at Metastatic Recurrence: A Case Report and Literature Review"

_diagnostics, 2025, doi:10.3390/diagnostics15233042_

Round 1

Reviewer 1 Report

Comments and Suggestions for Authors

This case report presents an interesting and clinically relevant case of ER loss and triple-negative conversion after adjuvant abemaciclib.

Title and Keywords
1- Keywords are few; add important terms such as “breast cancer relapse,” “triple-negative conversion,” or “case report.”

Abstract
2- Does not present key clinical data (e.g., diagnostic process, treatment course, outcomes) in detail.
3- Overly focused on background rather than patient details and case significance.

Introduction
4- Does not clearly state the rationale for presenting this particular case or its unique clinical relevance.
5- Missing a clear objective statement (why this case adds new knowledge).

Case report
6- lifestyle and relevant social/personal history are missing.
7- No explicit mention of patient concerns, symptoms, or expectations.
(CARE checklist recommends including patient perspective, which is absent here.)
8- Initial findings are summarized, but vital signs, general condition, and systemic examination results are not detailed.
9- No clear presentation of symptom evolution prior to diagnosis.
10- Does not describe physical findings at recurrence beyond “cutaneous nodules.”
11- Events are described narratively, but there is no visual timeline figure summarizing major diagnostic and treatment milestones. 
12- Please add a description of diagnostic reasoning, differential diagnoses, or decision-making process.
13- Laboratory results (routine bloodwork, biomarkers) not reported.
14- No mention of diagnostic challenges or limitations (e.g., differential diagnosis for cutaneous lesions).
15- NGS test is described. Please add its image.
16- The rationale for specific treatments (drug choices, sequence) is not clearly justified.
17- No mention of treatment-related adverse events, tolerance, or dose adjustments.
18- Lacks information on multidisciplinary decision-making process or patient involvement in therapy decisions.
19- Follow-up information is limited to short-term remission; duration of follow-up is not clearly specified.
20- No mention of ongoing surveillance, quality of life, or functional recovery.
21- Missing explicit description of final outcome at last contact (alive, disease-free, etc.).
22- No patient-reported outcomes or reflections included.

Discussion
23- Discussion of mechanisms (ER loss, PTEN, PI3K) dominates; connection to this patient’s findings is weak.
24- Missing section analyzing limitations of this case report (e.g., short follow-up, single case, lack of molecular correlation).
25- Please summarize lessons learned that were not explicitly provided.

Conclusion
26- Repeats literature points; does not summarize case-specific insights or implications for clinical practice.
27- Could be more focused on the main clinical message derived from this case.

Patient Perspective
Completely missing.

Figures and Tables
No clinical or radiologic figres of the patient are included in the text provided. (Please include representative images of key findings (e.g., imaging, pathology, biopsy), which are absent.

References
No comment.

Author Response

Thank you very much for taking the time to review this manuscript. Please find the corrections highlighted in the re-submitted files.

Reviewer 2 Report

Comments and Suggestions for Authors

The authors provide a case report on metastatic breast cancer and discuss the broad implication of the findings. the authors can include table for the patient's characteristics. could they provide more clarifications for the points below:

Which MD Anderson score was used? could you please include the links and parameters?
why was gBRCA1/2 uninformative? no known mutations detected? was it done for germline as well? 
Why did the patient not receive SERDs or trastuzumab initially? Did the TP53/PIK3CA mutations influence the treatment choices?

Author Response

Thank you very much for taking the time to review this manuscript. Please find the detailed responses below :

Which MD Anderson score was used? could you please include the links and parameters?

The score was calculated by the pathologist. No further information is available.

Why was gBRCA1/2 uninformative? no known mutations detected? was it done for germline as well? 

By “uninformative test,” I meant that the BRCA1 and BRCA2 genes were found to be wild-type.

Why did the patient not receive SERDs or trastuzumab initially? Did the TP53/PIK3CA mutations influence the treatment choices?

The patient did not receive a SERD because at disease recurrence she was triple-negative. She did not receive trastuzumab because she was HER2-low and not HER2-amplified.P53/PIK3CA mutations did not influence treatment choices but had only a prognostic significance in this case.

Round 2

Reviewer 1 Report

Comments and Suggestions for Authors

Dear Authors,
I accept the manuscript. It was improved significantly, and all ambiguities have been well resolved.